# Differential Association of Selected Adipocytokines, Adiponectin, Leptin, Resistin, Visfatin and Chemerin, with the Pathogenesis and Progression of Type 2 Diabetes Mellitus (T2DM) in the Asir Region of Saudi Arabia: A Case Control Study

**DOI:** 10.3390/jpm12050735

**Published:** 2022-05-01

**Authors:** Mohammad Muzaffar Mir, Rashid Mir, Mushabab Ayed Abdullah Alghamdi, Javed Iqbal Wani, Zia Ul Sabah, Mohammed Jeelani, Vijaya Marakala, Shahzada Khalid Sohail, Mohamed O’haj, Muffarah Hamid Alharthi, Mohannad Mohammad S. Alamri

**Affiliations:** 1Department of Basic Medical Sciences, College of Medicine, University of Bisha, Bisha 61922, Saudi Arabia; mjeelani@ub.edu.sa (M.J.); vijaya@ub.edu.sa (V.M.); sksohail@ub.edu.sa (S.K.S.); momohamed@ub.edu.sa (M.O.); 2Prince Fahd Bin Sultan Research Chair, Department of MLT, Faculty of Applied Medical Sciences, University of Tabuk, Tabuk 71491, Saudi Arabia; rashid@ut.edu.sa; 3Department of Internal Medicine, College of Medicine, University of Bisha, Bisha 61922, Saudi Arabia; mualghamdi@ub.edu.sa; 4Department of Internal Medicine, College of Medicine, King Khalid University, Abha 61421, Saudi Arabia; drjiwani1959@gmail.com (J.I.W.); drziaulsabah@gmail.com (Z.U.S.); 5Department of Family medicine, College of Medicine, University of Bisha, Bisha 61922, Saudi Arabia; mualharthi@ub.edu.sa (M.H.A.); malamri@ub.edu.sa (M.M.S.A.)

**Keywords:** adipocytokines, adipokines, adiponectin, leptin, resistin, visfatin, chemerin, T2DM pathogenesis, Asir Saudi Arabia

## Abstract

Background: Sedentary lifestyles, urbanization and improvements in socio-economic status have had serious effects on the burden of diabetes across the world. Diabetes is one of the 10 leading causes of death globally, and individuals with diabetes have a 2–3-fold increased risk of all-cause mortality. Adipose tissue is increasingly understood as a highly active endocrine gland that secretes many biologically active substances, including adipocytokines. However, the exact and discrete pathophysiological links between obesity and T2DM are not yet fully elucidated. Methods: In the current study, we present the association of five diverse adipocytokines, adiponectin, leptin, resistin, visfatin and chemerin, with T2DM in 87 patients (46 males and 41 females) with type 2 diabetes mellitus and 85 healthy controls (44 males and 41 females) from the Asir region of Saudi Arabia. The patients were divided into four groups: normal BMI, overweight, obese and severely obese. The baseline biochemical characteristics, including HbA1c and anthropometric lipid indices, such as BMI and waist–hip ratio, were determined by standard procedures, whereas the selected adipokine levels were assayed by ELISA. Results: The results showed significantly decreased levels of adiponectin in the T2DM patients compared to the control group, and the decrease was more pronounced in obese and severely obese T2DM patients. Serum leptin levels were significantly higher in the females compared to the males in the controls as well as all the four groups of T2DM patients. In the male T2DM patients, a progressive increase was observed in the leptin levels as the BMI increased, although these only reached significantly altered levels in the obese and severely obese patients. The serum leptin levels were significantly higher in the severely obese female patients compared to the controls, patients with normal BMI, and overweight patients. The leptin/adiponectin ratio was significantly higher in the obese and severely obese patients compared to the controls, patients with normal BMI, and overweight patients in both genders. The serum resistin levels did not show any significant differences between the males and females in thr controls or in the T2DM groups, irrespective of the BMI status of the T2DM patients. The visfatin levels did not reveal any significant gender-based differences, but significantly higher levels of visfatin were observed in the T2DM patients, irrespective of their level of obesity, although the higher values were observed in the obese and highly obese patients. Similarly, the serum chemerin levels in the controls, as well as in T2DM patients, did not show any significant gender-based differences. However, in the T2DM patients, the chemerin levels showed a progressive increase, with the increase in BMI reaching highly significant levels in the obese and severely obese patients, respectively. Conclusion: In summary, it is concluded that significantly altered concentrations of four adipokines, adiponectin, leptin, visfatin and chemerin, were found in the T2DM patient group compared to the controls, with more pronounced alterations observed in the obese and highly obese patients. Thus, it can be surmised that these four adipokines play a profound role in the onset, progression and associated complications of T2DM. In view of the relatively small sample size in our study, future prospective studies are needed on a large sample size to explore the in-depth relationship between adipokines and T2DM.

## 1. Introduction

Diabetes mellitus (DM) is a chronic disease and is a major global healthcare problem, affecting both public health and socio-economic wellbeing. The prevalence of the disease has increased gradually in recent times in both developing and developed countries, although the incidence has shown a decline in some countries [1]. The International Diabetes Federation (IDF) has projected an increase in the number of diabetics to 693 million by 2045 [2]. Diabetes is one of the top 10 causes of death globally and individuals with diabetes have a 2–3-fold increased risk of all-cause mortality [1]. Saudi Arabia, one of the countries in the MENA region, has a diabetes prevalence of 17.7%, with the total number of adult diabetics equaling 4,274,100 at the time of writing (https://idf.org/our-network/regions-members/middle-east-and-north-africa/members/46-saudi-arabia.html (accessed on 13 March 2022). DM is associated with acute consequences, such as diabetic ketoacidosis and hyperosmolar hyperglycemic syndrome, as well as chronic complications, such as renal failure, blindness, cardiovascular disease and diabetic neuropathy [3]. In general, there are two types of DM; type 1 DM (T1DM) is caused by the destruction of the beta cells of the pancreas, which secrete insulin [4], while type 2 DM (T2DM) develops through tissue resistance to insulin and pancreatic beta-cell dysfunction [4]. Lifestyle changes, including urbanization, the increasing pace of life, the consumption of high-calorie diets and lack of physical activity have resulted in a high burden of obesity and concomitant diabetes [5]. Obesity is regarded as one of the main risk factors for diabetes, and about 90% of diabetics are estimated to be obese or overweight [6,7].

There is significant evidence that suggests a connection between the increased lipid concentration in the cytoplasm of adipocytes, myocytes and hepatocytes and the development of insulin resistance in peripheral tissues [7,8,9]. Additionally, the common therapies used to treat T2DM (e.g., sulphonyl urea derivatives, thiazolidinediones and insulin) may, in addition, lead to weight gain and subsequent insulin resistance [6]. The pathophysiology of insulin resistance and T2DM has been a focus of many studies [7]. A plausible assumption is that T2DM is partly increased by the altered functions of adipose tissue [10]. Adipose tissue is increasingly understood as a highly active endocrine gland that secretes many biologically active substances, including adipocytokines [7,11,12]. However, the exact and discrete pathophysiological links between obesity and T2DM are not yet fully elucidated. There is enough evidence to suggest that changes in adipocytokine secretion do contribute to defective insulin production/action and, concomitantly, result in peripheral insulin resistance [13,14,15,16,17]. Adipokines such as adiponectin, leptin resistin, visfatin, ghrelin and chemerin seem to play an important role in the onset, progression and complications of T2DM [14,17,18,19,20,21,22].

Adiponectin is one of the most interesting and abundant adipocytokines, and it has been reported to improve insulin resistance and inflammatory status [23,24,25]. This function of adiponectin is potentially performed through its role in increased fatty acid oxidation in the liver and skeletal muscles, as well as through its inhibition of gluconeogenesis in liver [26]. Decreased levels of adiponectin have been associated with obesity-related diseases, including insulin resistance, type 2 diabetes mellitus, and cardiovascular disease [27,28,29]. Many studies have associated adiponectin with a low risk of T2DM and, in fact, its levels are shown to be reduced before diabetes onset [30,31]. Another study revealed that adiponectin levels are lower in prediabetes than in a euglycemic state [32] Adiponectin levels were related to impaired glucose tolerance in another report [33].

Leptin, also known as an “anorexigenic” hormone, is another adipokine that is synthesized in adipose tissues and enterocytes and mediates the long-term regulation of energy balance and suppressing food intake, thereby inducing weight loss [5,34]. On one hand, leptin increases lipolysis and fatty acid oxidation while, on the other hand, it decreases lipogenesis [34]. Apart from obesity, hyperleptinemia has also been associated with hypertension and insulin resistance [35,36], while chronic hyperlipidemia is believed to exert deleterious effects on the functions of leptin [37].

Resistin, also known as adipose-tissue-specific secretory factor (ADSF), is another adipokine with a 108-amino acid polypeptide synthesized in adipocytes, pancreatic cells, muscle, and peripheral-blood mononuclear cells [38,39,40]. It is believed to play a significant role in the development of insulin resistance [41]. Although the role of resistin in pro-inflammatory processes has been demonstrated in several studies [42,43], the role of resistin in insulin resistance in humans remains controversial. While some studies reported positive correlations between resistin and insulin resistance in T2DM [44,45,46,47], others failed to establish such an association [48,49].

Visfatin is a multifaceted novel adipokine. It is also known as nicotinamide phosphoribosyl transferase (NAMPT) or pre-B cell colony-enhancing factor (PBEF). It is preferentially expressed in visceral adipose tissue, although it is also present in the cytoplasm and the nucleus of cells in many tissues and organs, including the kidneys, testes, brain, lung and spleen [50]. It has been found to have a vast array of endocrine and autocrine paracrine functions that include cell proliferation, the biosynthesis of nicotinamide mono and dinucleotide and hypoglycemic effects [50,51]. The increased expression and plasma levels of visfatin are associated with T2DM and abdominal obesity [52]. The hypoglycemic actions of visfatin include the inhibition of the release of glucose from hepatic cells and the activation of glucose uptake in peripheral tissues by binding insulin receptors at a site different from that of insulin [52]. There are contradictory reports about the role of visfatin in insulin resistance and diabetes: some reports suggest a link with visfatin [52,53,54], while others suggest that visfatin is not independently related to diabetes [55,56].

Chemerin is a fairly new adipokine that has been found to perform endocrine, paracrine and autocrine functions and plays a vital role in lipid and glucose homeostasis [57]. In humans, it is mainly produced by white adipose tissue, the liver and the placenta and, in minor amounts, by brown adipose tissue, the skeletal muscles, the kidneys, the adrenals, the lungs, the ovaries and the heart [57,58,59,60]. Chemerin is also involved in angiogenesis, inflammation, immune modulation and the regulation of blood pressure [61]. Many researchers have reported higher levels of chemerin in diabetic patients compared to healthy controls, whereas others have reported no significant differences in chemerin levels in patients with diabetes and healthy controls [61,62].

In view of the contradictory reports about the involvement of selected adipokines, this case-control and exploratory study was designed to examine the association of five diverse adipokines, namely, adiponectin, leptin, resistin, visfatin and chemerin with T2DM in a section of patients from the Asir region of Saudi Arabia.

## 2. Methodology

Study population: This case-control and collaborative study was conducted on 87 T2DM patients and 85 healthy controls. Blood specimens were collected from Asir region of Saudi Arabia from hospitals of Bisha and Abha cities. The recruitment period of the patients and controls was from March 2020 to January 2021. Prior to the collection of blood samples, informed consent was obtained from all patients and control subjects. 

Ethical Approval: Ethical approval was obtained from local RELOC committee of the College of Medicine, University of Bisha (ref. no. UBCOM/H-06-BH-087(04/10), in accordance with the local guidelines, which are aligned, in essence, with the principles of the Helsinki Declaration. 

Inclusion criteria: 

In this study, only Saudi citizens were enrolled. Initially, 100 patients were included who had fasting plasma glucose level > 110 mg/dL and/or who were clinically confirmed as being on oral hypoglycemic agents or insulin and had fasting glucose levels ≤ 110 mg/dL on the day of blood sampling. Patients with random blood glucose greater than 200 mg/dL and/or those clinically confirmed as being on oral hypoglycemic agents or insulin and had random glucose levels less than 200 mg/dL on the day of blood sampling were also included. In total, 87 T2DM patients were included in this study, which included 46 males and 41 females. 

Exclusion Criteria: 

T2DM patients with other significant chronic diseases or malignancies were excluded from the study. Type 1 diabetes patients were also excluded from the study. 

Inclusion criteria for controls: Initially, about 100 control subjects who were healthy volunteers with no previous record of diabetes or any major clinical disorders and had normal fasting and random plasma glucose levels were included.

Exclusion criteria for controls: The control subjects whose investigations revealed abnormal results were excluded from the study, and only 85 control subjects (44 males and 41 females) were included in the study.

Data collection: This case-control study included Saudi citizens who were patients the hospitals in Bisha and Abha regions and clinically confirmed as having T2DM. The study enrolled 87 subjects with T2DM and 85 normal control subjects. T2DM was diagnosed according to the parameters of WHO criteria. The specimen collection was performed at Diabetic Center, King Abdullah Hospital, Bisha and at Asir General Hospital, Abha. The recruitment period of the patients and controls was from March 2020 to January 2021. However, blood sample collection was not performed during the month of “Ramadhan”. The various variables that were analyzed from the T2DM patients and controls included the case history, age, gender, waist-to-hip ratio (WHR), body-mass index (BMI), glycated hemoglobin (HbA1c), fasting and random blood glucose levels, total cholesterol, triacylglycerol (TG), high-density lipoprotein-cholesterol (HDL-C) and low-density lipoprotein cholesterol (LDL-C) concentrations. The biochemical parameters were assayed using the standard protocols. 

Sample collection from T2DM patients: About 4 mL of peripheral blood sample was collected in a red top tube (without any anticoagulant) from all T2DM patients. The serum was separated and one serum aliquot was immediately stored at −20 °C to −30 °C until further studies were conducted. Another serum aliquot was immediately sent for biochemical analyses. 

Sample collection from healthy controls: Sample collections from all healthy age-matched control subjects were timed around routine blood draws that were part of the routine health checkup and, hence, did not require additional phlebotomy, and all participants were provided with a written informed consent form. About 4 mL of peripheral blood sample was collected in a red top tube (without any anticoagulant) for all controls. The serum was separated and one serum aliquot was immediately stored at −20 °C to −30 °C until further studies were conducted. Another serum aliquot was immediately sent for biochemical analyses. 

Estimation of Adipocytokines: The serum concentrations of the studied adipokines were determined by enzyme-linked immunosorbent assays (ELISA) using commercially available kits from ABCAM, USA, and following the instructions of the manufacturer. The intra- and inter-assay variations were less than 10%.

Adiponectin: Adiponectin was estimated by using human adiponectin ELISA kit, catalogue number ab222508, and the levels were expressed as µg/mL.

Leptin: Leptin concentration was determined by using human leptin ELISA kit, catalogue number ab179884, and the levels were expressed as ng/mL.

Resistin: Resistin levels were estimated by using human resistin ELISA kit, catalogue number ab183364, and the levels were expressed a ng/mL.

Visfatin: Visfatin levels were estimated by using human visfatin ELISA kit, catalogue number ab264623, and the levels were expressed as ng/mL.

Chemerin: Chemerin concentration was determined by using human chemerin ELISA kit, catalogue number ab155430, and the levels were expressed as ng/mL.

Statistical analysis: The statistical analysis was performed by using SPSS version 20 software. Normally distributed data were expressed as mean with standard deviation (SD), and data with skewed distribution were shown as medians (Q1–Q3). For variables with normal distribution and homogenous variance differences, significances were determined using a one-way analysis of variance (ANOVA) followed by Tukey HSD test, otherwise Kruskal–Wallis one-way analysis of variance by ranks together with multiple-comparison post hoc test was applied. Values were considered significant with *p* < 0.05. The graphs were created by using R programming language (www.R-project.org, accessed on 9 April 2022). 

## 3. Results

Characteristics of male T2DM patients: The anthropometric indices of male T2DM patients and controls are summarized in Table 1.

The 46 male T2DM patients were divided into 4 groups on the basis of their BMI values. Group A had 12 patients with normal body weight and BMI of 21.78 ± 2.10 kg/m^2^; group B included 11 overweight patients with BMI of 27.26 ± 2.28 kg/m^2^; group C had 13 patients with obesity and a BMI of 34.87 ± 3.16 kg/m^2^; and group D included 10 T2DM patients with severe obesity and a BMI of 46.54 ± 5.68 kg/m^2^. The 44 male control subjects had a normal BMI of 21.74 ± 1.78 kg/m^2^, as presented in Table 1. The ages of the male T2DM patients ranged from 28 to 65 years and those of the controls ranged from 28 to 62 years. The WHR was significantly higher in the group C and group D patients compared to the controls. Group D showed a highly significant increase in WHR *p* < 0.001 compared to groups A and B, whereas group C patients showed a significant increase, with *p* < 0.01 in the WHR versus groups A and B. The BMI levels showed no significant increase in groups A and B, whereas they were significantly increased in groups C and D compared to the controls. The group C patients had significantly increased BMI compared to those in groups A and B, with *p* < 0.0.01, whereas the group D patients had significantly increased BMI compared with those in groups A and B (*p* < 0.001) and significantly increased BMI compared with those in group C, with *p* < 0.01.

The biochemical parameters of the male T2DM patients and the controls are depicted in Table 1. The fasting levels of glucose did not show any significant differences in the four groups compared to the controls.

HbA1c was significantly increased in all the four groups of the T2DM patients compared to the control group, with *p* < 0.001, but did not show any significant difference among the different patient groups, A–D. The lipid indices in all four groups did not show any significant difference compared to the controls.

Characteristics of female T2DM patients: The anthropometric indices of the female T2DM patients and controls are summarized in Table 2.

The 42 female T2DM patients were also divided into 4 groups on the basis of their BMI values. Group A had 11 patients with normal body weight and a BMI of 21.32 ± 2.18 kg/m^2^; group B included 10 overweight patients with BMI of 28.42 ± 2.32 kg/m^2^; group C had 11 patients with obesity and a BMI of 33.98 ± 3.22 kg/m^2^; and group D included 9 T2DM patients with severe obesity and a BMI of 47.22 ± 4.88 kg/m^2^. The 41 female control subjects had a normal BMI of 21.20 ± 1.66 kg/m^2^, as can be seen in Table 2. The ages of the female T2DM patients ranged from 27 to 65 years and those of the controls ranged from 27 to 64 years. The WHR was significantly higher in the group C and group D patients compared to the controls. Group D showed a highly significant increase in WHR, with *p* < 0.001, compared to groups A and B, whereas the group C patients showed a significant increase, *p* < 0.01, in WHR versus those in groups A and B. The BMI showed no significant increase in groups A and B, whereas it was significantly increased in groups C and D compared to the controls. The group C patients had significantly increased BMI compared to groups A and B, with *p* < 0.0.01, whereas the group D patients had significantly increased BMI compared with those in groups A and B (*p* < 0.001) and significantly increased BMI compared with those in group C, with *p* < 0.01. The biochemical parameters of the female T2DM patients and controls are depicted in Table 2. The fasting levels of glucose did not show any significant differences in the four groups compared to the controls. The HbA1c was significantly increased in all four groups of the T2DM patients compared to the control group, with *p* < 0.001, but did not show any significant difference among the different patient groups, A–D. The lipid indices in all four groups did not show any significant difference compared to controls.

Concentrations of adipokines in study groups

Adiponectin:

As can be seen in Table 3, the concentration of adiponectin was significantly higher in the females (19.62 ± 4.72 µg/mL) compared to the males (14.22 ± 3.21 µg/mL), with *p* < 0.01 in the control group. In both groups C and D of T2DM patients (males and females), the adiponectin levels were significantly lower *p* < 0.01 compared to the control subjects. In the males, groups C and D also showed a significant decrease in adiponectin levels relative to group A. However, among four female T2DM patient groups, no difference was found in the adiponectin levels. A graphic representation of the adiponectin levels is shown in Figure 1.

Leptin: In the control group, the leptin levels were found to be significantly higher in the females (25.42 ± 5.78 * ng/mL) compared to the males (5.42 ± 1.98 ng/mL) *p* < 0.001). In the women, the plasma leptin concentrations were significantly increased in the group D patients with severe obesity compared to the controls. However, the men with T2DM with obesity in group C presented significantly high levels of leptin compared to the controls and groups A and B, with *p* < 0.01, whereas the men with T2DM in group D with severe obesity demonstrated significantly high levels of adiponectin compared to the controls and groups A and B, with *p* < 0.001. The leptin levels are summarized in Table 1 and graphically depicted in Figure 2.

Leptin–adiponectin ratio: As can be seen in Table 3, the leptin–adiponectin ratio in the males was significantly higher in group C compared to the controls and group A with, *p* < 0.01, whereas group D showed a highly significant increase in this ratio compared to the controls and groups A–C, with *p* < 0.001. The leptin/adiponectin ratio in the females showed a gradual increase in all the T2DM groups compared to the controls, but it was significant only in group C (*p* < 0.01) and group D, with *p* < 0.001. Although there was an increase in the adiponectin–leptin ratio in the female groups A and B compared to the controls, it was not significant.

Resistin: The serum levels of resistin, visfatin and chemerin are summarized in Table 4. The serum resistin levels did not show any significant difference between the males (22.86 ± 4.88 ng/mL) and females (24.32 ± 4.94 ng/mL) in the control group. Neither the males nor the females in the four T2DM patient groups demonstrated significant differences, as can be seen in Table 4 and Figure 3.

Visfatin: The results for serum visfatin are also depicted in Table 4 and graphically represented in Figure 4, where it can be seen that the concentrations of visfatin were not significantly different between the males (3.82 ± 1.74 ng/mL) and females (3.16 ± 1.62 ng/mL) in the control group; however, the T2DM patients in all four groups, irrespective of gender, showed an increase in visfatin levels. In the males, the T2DM patients ingroups A and B showed significant differences in visfatin levels compared to the controls, with *p* < 0.05, whereas the T2DM patients in groups C and D in T2D showed highly significant differences in visfatin levels compared to the controls, with *p* < 0.01. In the females, the T2DM patients in groups B and D showed significant differences in their visfatin levels compared to the controls, with *p* < 0.05, whereas the female T2DM patients in group C showed highly significant differences in visfatin levels compared to the controls, with *p* < 0.01. In the females, the group C T2DM patients showed a significantly high level of visfatin (8.12 ± 2.32 ng/mL) compared to those in group A (5.44 ± 1.86 ng/mL), with *p* < 0.05. On the other hand, no significant differences were found among the male patients with T2DM in groups A to D.

Chemerin: The levels of chemerin in the T2DM patients and controls are summarized in Table 4 and graphically represented in Figure 5. As is evident, the levels of chemerin were not significantly different between the males (268.42 ng/mL) and females (262.13 ng/mL) in the control group. In the T2DM patients, the chemerin levels showed a steady increase in all four groups, A to D, compared to the controls. However, both the male and the female T2DM patients with obesity in group C showed significant elevation in their chemerin concentrations, with *p* < 0.05, whereas both the male and the female patients with severe obesity in group D showed significantly elevated levels of chemerin, with *p* < 0.01. There was no significant gender-based difference in chemerin levels in any of the four groups, although the females showed a lower trend in all the groups.

## 4. Discussion

Diabetes has assumed epidemic proportions and, according to the International Diabetes Federation, one in eight individuals will have this disease by 2045 [1]. The prevalence is higher in urban than in rural areas and in high-income nations than in those with low income [1,2]. The roles of adipose tissue and adipocytokines in insulin resistance and alterations in lipid metabolism are becoming increasingly clear. In this study, we determined the levels of five selected adipokines in four groups of male and female patients with T2DM patients compared to age-matched healthy controls.

The results showed significantly decreased levels of adiponectin in the T2DM patients compared to the control group, which is in agreement with the results of earlier studies [27,28,29]. The decrease was more pronounced in the obese and severely obese T2DM patients, which corroborates the results of earlier reports, which showed significant decreases in adiponectin levels in overweight and obese diabetics [23,24,25]

Although the adiponectin levels decreased significantly in the female T2DM patients compared to the controls, the decrease was not significant among the four groups of female T2DM patients. In our study groups, hypoadiponectinemia seemed to be inversely proportional to the level of obesity, which was in agreement with earlier studies. The effect of adiponectin on insulin sensitivity conceivably involves an increase in the phosphorylation and the activation of AMPK in the skeletal muscles and liver by adiponectin, leading to improved muscle–fat oxidation and glucose transport. Since adiponectin is an insulin-sensitizing adipokine and exerts its effects through AMP-activated protein kinase (AMPK), its profound decrease in obese and severely obese individuals makes them more susceptible to T2DM. [27,28,29,63,64].

The serum leptin levels were significantly higher in the females compared to the males in the controls, as well as all the four groups of T2DM patients. In the male T2DM patients, a progressive increase was observed in the leptin levels as the BMI increased, reaching significantly altered levels, although this was only observed in the obese and severely obese patients. The serum leptin levels were significantly higher in the severely obese female patients (group D) compared to the controls and groups A and B. In the group C females, a slight but insignificant increase was observed in the leptin levels compared to the controls and groups A and B. These results are in line with those of earlier reports [8,9]. The higher corresponding leptin levels in the females may be explained on the basis of the sexual dimorphism of adipose tissue distribution, as females mostly accumulate fat subcutaneously, which leads to higher levels of leptin, which are augmented by 17β-estradiol-dependent secretions of leptin [5,65].

We also studied the leptin–adiponectin ratio, as it is thought to be a better determinant of insulin resistance and metabolic alterations than leptin or adiponectin levels alone. When a comparison was drawn with the degree of obesity, it was observed that the leptin–adiponectin ratio was significantly higher in the female, obese T2DM patients compared to the controls, whereas in the males it was higher compared to both the controls and the diabetics with normal body weight, which was consistent with other studies [28,29,35]

The results on the serum resistin levels did not show any significant differences between the males and females in the controls or the T2DM groups, irrespective of the BMI status of the T2DM patients. Our results are in agreement with those of earlier reports [48,49]. Although a few studies have reported a positive correlation between resistin levels and insulin resistance [44,45,46,47], this issue needs to be addressed in larger human studies in different ethnic populations.

The results on the visfatin levels did not reveal any significant gender-based differences, which is consistent with earlier observations [53,54]. Significantly higher levels of visfatin were observed in the T2DM patients, irrespective of their level of obesity, although higher values were observed in the obese and highly obese patients, as reported previously (56,57]. Several reports suggest a positive correlation of visfatin with insulin resistance [52,54] whereas others do not suggest a direct role for visfatin in insulin resistance [55,56]. Fioravanti et al. reported that visfatin levels were substantially reduced in non-diabetic individuals after three weeks of weight-loss therapy, but the reverse was observed with regard to the serum visfatin levels in T2DM patients [66]. This can be explained on the basis of the hypothesis that such a phenomenon developed as a compensatory response to impaired insulin action, thereby confirming that the plasma visfatin levels were a consequence of the degree of insulin resistance. Nevertheless, the exact relationship between serum visfatin levels and insulin resistance remains unclear, and relevant studies have reported conflicting results [53,54,55,56].

The chemerin levels in both the controls and the T2DM patients did not show any significant gender-based differences, as in the case of the visfatin. However, in the T2DM patients, the chemerin levels showed a progressive increase in line with increases in BMI, reaching highly significant levels in the obese and severely obese patients, respectively. Our results are corroborated by earlier and recent findings [62,67]. In the current study, higher levels of chemerin were observed in the T2DM patients and were closely associated increasing WHR and BMI, which is in agreement with the work of Yang et al., who reported a positive correlation between chemerin levels and BMI and other anthropometric indices [68]. Our results are also in agreement with a previous study conducted on a section of Saudi women by Habib et al., who reported higher levels of chemerin in T2DM patients, and a good correlation with insulin resistance and adiposity [69]. Taken together, these results suggest a strong role of chemerin in the pathogenesis of insulin resistance, which is further compounded by obesity.

## 5. Conclusions

In this study, the associations of the serum levels of five selected adipokines as a function of BMI and WHR in four groups of male and female T2DM patients compared to age-matched healthy controls were investigated.

The findings of this research were as follows:⇒Significantly decreased levels of serum adiponectin were found in the T2DM patients compared to the control group, with the decrease being more pronounced in the obese and severely obese T2DM patients.⇒Significantly higher serum leptin was found in the females compared to the males in the controls as well as all four groups of T2DM patients. In the male T2DM patients, a progressive increase was observed in leptin levels with increasing BMI, although this only reached significantly altered levels in the obese and severely obese patients. The serum leptin levels were also significantly higher in the severely obese female patients compared to the controls, patients with normal BMI and overweight patients.⇒The leptin–adiponectin ratio was significantly higher in the obese and severely obese patients compared to the controls, patients with normal BMI, and overweight patients in both genders.⇒No significant differences were noticed in the serum resistin levels between the males and females in the controls or in the the T2DM groups, irrespective of the BMI status of the T2DM patients.⇒No significant gender-based differences were identified in the visfatin levels; however, significantly higher levels of visfatin were observed in the T2DM patients irrespective of their level of obesity, although the highest values were observed in the obese and highly obese patients.⇒No significant gender-based differences were recorded in the serum chemerin levels in the controls or in the T2DM patients, but in the T2DM patients, the chemerin levels showed a progressive increase in line with the increase in BMI, reaching highly significant levels in the obese and severely obese patients, respectively.

Altogether, it is concluded that the altered secretions of four adipokines, adiponectin, leptin, visfatin and chemerin, may contribute to the defective production/action of insulin and, concomitantly, insulin resistance. Thus, it can be surmised that these four adipokines play a profound role in the onset, progression and associated complications of T2DM. In view of the relatively small sample size—a possible limitation of our study—these observations cannot be generalized to all T2DM patients. We recommend more prospective investigations on large sample sizes and in different ethnic populations to explore the in-depth relationship between adipokines and T2DM.

## Figures and Tables

**Figure 1 jpm-12-00735-f001:**
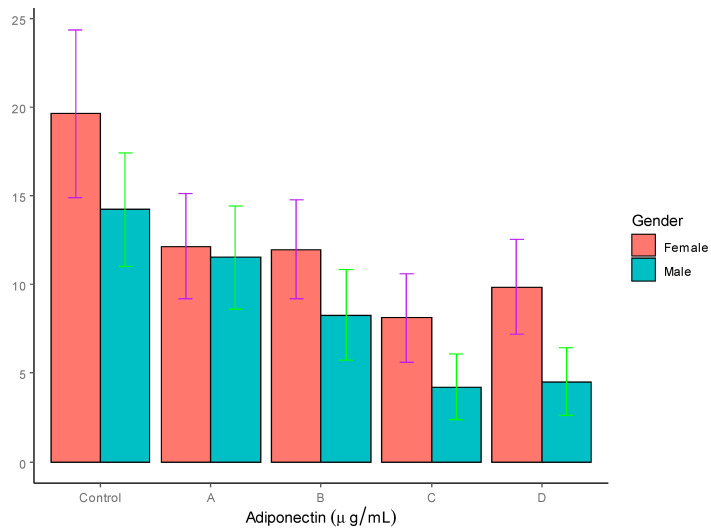
Serum adiponectin levels in T2DM patients and controls (µg/mL).

**Figure 2 jpm-12-00735-f002:**
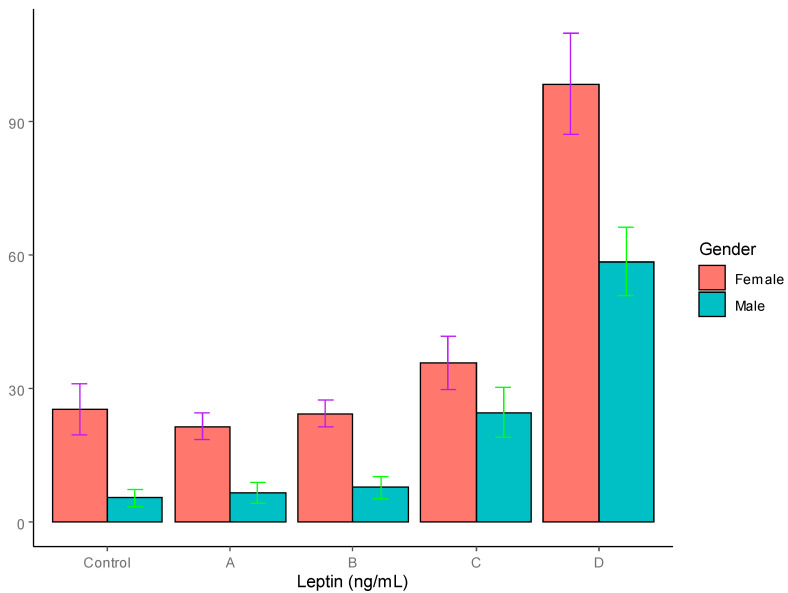
Serum leptin levels in T2DM patients and control groups (ng/mL).

**Figure 3 jpm-12-00735-f003:**
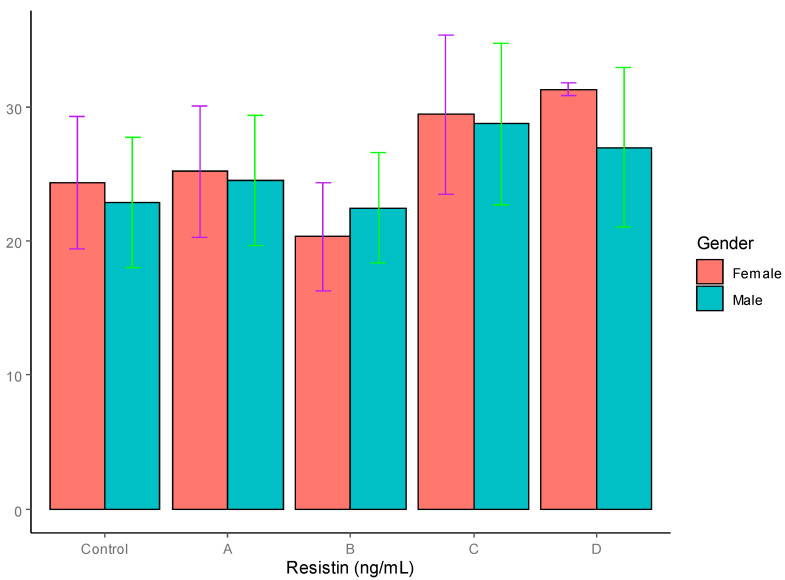
Serum resistin levels in T2DM patients and control groups (ng/mL).

**Figure 4 jpm-12-00735-f004:**
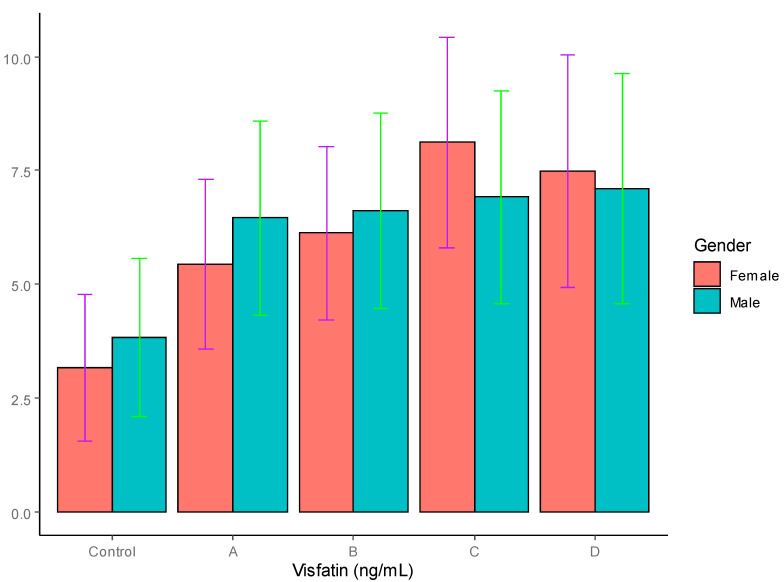
Serum visfatin levels in T2DM patients and control groups (ng/mL).

**Figure 5 jpm-12-00735-f005:**
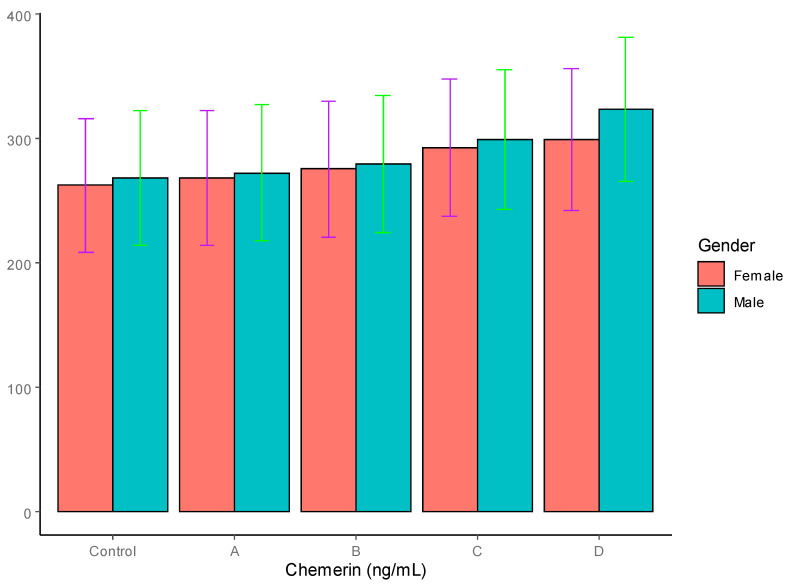
Serum adiponectin levels in T2DM patients and control groups (ng/mL).

**Table 1 jpm-12-00735-t001:** (**a**) Anthropometric indices of male T2DM patients and controls. (**b**) Biochemical parameters of male T2DM patients and controls.

**(a)**						
		**Group A**	**Group B**	**Group C**	**Group D**	**Controls**
	**Number**	**12**	**11**	**13**	**10**	**44**
**Parameter**						
Age in years (range)		44 (29–52)	46 (30–54)	49 (29–63)	46 (33–65)	46 (28–62)
WHR		0.91 (0.85–0.1.04)	0.92 (0.84–0.99)	1.06 ** ^N^ (0.98–1.10)	1.12 * ^M^ (1.04–1.18)	0.86 (0.84–0.95)
BMI (kg/m^2^)		21.78 ± 2.10	27.26 ± 2.28	34.87 ± 3.16 * ^N^	46.54 ± 5.68 * ^XY^	21.78 ± 1.78
**(b)**						
		**Group A**	**Group B**	**Group C**	**Group D**	**Controls**
	**Number**	**12**	**11**	**13**	**10**	**44**
**Parameter**						
Fasting glucose (mg/dL)		112 (88–128)	108 (92–135)	113 (94–142)	118 (101–146)	92 (78–116)
HbA1c (g/dL)		7.5 ± 1.01 *	7.9 ± 1.2 *	8.6 ± 1.46 * ^M^	7.4 ± 0.98 *	4.9 ± 0.88
Cholesterol-T		206 (155–234)	209 (142–226)	211 (144–238)	216 (153–246)	186 (135–224)
HDL-C (mg/dL)		48 (36–62)	52 (38–58)	47 (34–60)	44 (29–55)	53 (38–63)
LDL-C (mg/dL)		94 ± 32.20	110 ± 29.42	117 ± 36.32	134 ± 32.34	94 ± 32.20
TG (mg/dL)		116 (101–189)	146 (98–220)	145 (95–202)	215 (112–228)	104 (82–112)

For Part (a): WHR—waist-to-hip ratio; BMI—body mass index. Group A—T2DM patients with normal body weight; group B–overweight T2DM patients; group C—T2DM patients with obesity and group D —T2DM patients with severe obesity. * *p* < 0.001 vs. control group; ** *p* < 0.01 vs. control group; ^M^ *p* < 0.001 vs. groups A and B, ^N^ *p* < 0.01 vs. groups A and B; ^X^ *p* < 0.001 vs. group A; ^Y^ *p* < 0.01 vs. group C. For Part (b): HbA1c—glycated hemoglobin; Cholesterol-T—total cholesterol; HDL—high-density lipoprotein; LDL—low-density lipoprotein; TG—triglycerides. Group A—T2DM patients with normal body weight; group B—overweight T2DM patients; group C—T2DM patients with obesity; and group D —T2DM patients with severe obesity. * *p* < 0.001 vs. control group; ^M^ *p* < 0.001 vs. group A and B.

**Table 2 jpm-12-00735-t002:** (**a**) Anthropometric indices of female T2DM patients and controls. (**b**) Biochemical parameters of female T2DM patients and controls.

**(a)**						
		**Group A**	**Group B**	**Group C**	**Group D**	**Controls**
	**Number**	**11**	**10**	**11**	**9**	**41**
**Parameter**						
Age		45 (27–56)	48 (29–55)	47 (31–65)	48 (29–64)	48 (27–64)
WHR		0.94 (0.84–1.03)	0.95 (0.82–1.02)	1.05 (0.98–1.12) ** ^N^	1.08 (1.01–1.16) * ^M^	0.85 (0.78–0.96)
BMI (kg/m^2^)		21.32 ± 2.18	28.42 ± 2.32	33.98 ± 3.22 * ^N^	47.22 ± 4.88 * ^XY^	21.2 ± 1.66
**(b)**						
		**Group A**	**Group B**	**Group C**	**Group D**	**Controls**
	**Number**	**11**	**10**	**11**	**9**	**41**
**Parameter**						
Fasting glucose(mg/dL)		117 (92–131)	111 (87–129)	112 (92–139)	121 (97–152)	96 (82–114)
HbA1c (g/dL)		7.9 ± 1.14 *	8.1 ± 1.08 *	8.2 ± 1.38 * ^M^	8.1 ± 0.89 *	4.5 ± 0.43
Cholesterol-T		198 (149–229)	208 (145–240)	216 (156–232)	223 (161–253)	168 (132–224)
HDL-C(mg/dL)		52 (41–63)	50 (40–60)	44 (33–61)	42 (31–57)	56 (41–66)
LDL-C(mg/dL)		88 ± 26.34	108 ± 32.12	103 ± 29.28	121 ± 28.18	96 ± 28.22
TG (mg/dL)		120 (98–172)	138 (90–218)	146 (102–225)	186 (101–238)	96 (88–124)

For Part (a): WHR—waist-to-hip ratio; BMI—body mass index. Group A—T2DM patients with normal body weight; group B–overweight T2DM patients; group C—T2DM patients with obesity; and group D —T2DM patients with severe obesity. * *p* < 0.001 vs. control group; ** *p* < 0.01 vs. control group; ^M^ *p* < 0.001 vs. groups A and B, ^N^ *p* < 0.01 vs. groups A and B; ^X^ *p* < 0.001 vs. group A; ^Y^ *p* < 0.01 vs. group C. For Part (b): HbA1c—glycated hemoglobin; Cholesterol-T—total cholesterol; HDL—high-density lipoprotein; LDL—low-density lipoprotein; TG—triglycerides. Group A—T2DM patients with normal body weight; group B—overweight T2DM patients; group C—T2DM patients with obesity; and group D —T2DM patients with severe obesity. * *p* < *0*.001 vs. control group; ^M^ *p* < 0.001 vs. group A and B.

**Table 3 jpm-12-00735-t003:** Serum adiponectin and leptin concentration in T2DM patients and controls.

		Adiponectin (µg/mL)		Leptin (ng/mL)		Leptin:Adiponectin Ratio	
	Gender	Males	Females	Males	Females	Males	Females
Groups							
Controls		14.22 ± 3.21	19.62 ± 4.72 *	5.42 ± 1.98	25.42 ± 5.78	0.38	1.29
A		11.53 ± 2.90	12.15 ± 2.96	6.51 ± 2.32	21.44 ± 3.01	0.56	1.76
B		8.26 ± 2.55	11.96 ± 2.79	7.78 ± 2.45	24.35 ± 2.99	0.94	2.04
C		4.21 ± 1.85 *	8.12 ± 2.48 *	24.65 ± 5.62 *	35.78 ± 6.05	5.85	4.41
D		4.53 ± 1.92 *	9.85 ± 2.67 *	58.62 ± 7.82 **	98.60 ± 11.34 ** ^M^	12.95 ** ^M^	10.01 ** ^M^

Group A—T2DM patients with normal body weight; group B—overweight T2DM patients; group C—T2DM patients with obesity and group D—T2DM patients with severe obesity. * *p* < 0.01 vs. control group; ** *p* < 0.001 vs. control group; ^M^ *p* < 0.04 vs. group C.

**Table 4 jpm-12-00735-t004:** Serum resistin, visfatin and chemerin concentrations and their ratios in T2DM patients and controls.

		Resistin(ng/mL)		Visfatin(ng/mL)		Chemerin (ng/mL)	
	Gender		Females	Males	Females	Males	Females
Groups							
Controls		22.86 ± 4.88	24.32 ± 4.94	3.82 ± 1.74	3.16 ± 1.62	268.42 ± 54.23	262.13 ± 54.03
A		24.54 ± 4.85	25.21 ± 4.90	6.45 ± 2.13 *	5.44 ± 1.86	272.12 ± 54.69	267.72 ± 54.11
B		22.45 ± 4.12	20.32 ± 4.04	6.62 ± 2.15 *	6.12 ± 1.91	279.35 ± 55.12	275.32 ± 54.89
C		28.76 ± 6.04	29.43 ± 5.98	6.92 ± 2.34 **	8.12 ± 2.32 ***^#^	298.65 ± 56.34 *	292.45 ± 55.12 *
D		26.98 ± 5.94	31.32 ± 0.45	7.1 ± 2.54 **	7.48 ± 2.56 **^#^	323.43 ± 58.23 **^#^	299.12 ± 57.03 **

Group A—T2DM patients with normal body weight; group B—overweight T2DM patients; group C—T2DM patients with obesity and group D—T2DM patients with severe obesity. * *p* < 0.05 vs. controls; ** *p* < 0.01 vs. controls; *** *p* < 0.001 vs. controls, ^#^
*p* < 0.05 vs. groups A and B.

## Data Availability

Data used in this study is available upon reasonable request.

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
