# Peer review of "Differential Association of Selected Adipocytokines, Adiponectin, Leptin, Resistin, Visfatin and Chemerin, with the Pathogenesis and Progression of Type 2 Diabetes Mellitus (T2DM) in the Asir Region of Saudi Arabia: A Case Control Study"

_jpm, 2022, doi:10.3390/jpm12050735_

Round 1

Reviewer 1 Report

The manuscript entitled " Differential association of selected adipocytokines; adiponectin, leptin, resistin, visfatin, and chemerin, with the pathogenesis and progression of type 2 diabetes mellitus (T2DM) in Asir region of Saudi Arabia"  is focused on the measurement of 4 adipocytokines in patients diagnosticated with T2DM. Authors may find relevant suggested changes and recommendations:

  1. The authors must describe in more detail what is new in this manuscript. Please indicate how your work is different and important versus other studies published in the literature.
  2. What are the advantages, novelty, or new contributions to the knowledge compared with other works reported previously in literature? Please list it in a table and explain.
  3. How do you select the N (number of participants) in each group? describe the methodology in the proper section.  
  4. Table 1 is difficult to follow. I suggest splitting the table into 2 tables 1 about the biochemical parameters, and a second table about the demographic and anthropometric parameters.
  5. Table 2 is difficult to follow too and is difficult to compare with Table 1. Please, be sure to rearrange the information so that it can be compared (men and women) and divide it into the characteristics that are related.
  6. Table 3 should be a Figure in order to make it easier to follow and compare. 
  7. The colors are not suitable for a science project. please use more formal but different colors that allow treatments to be compared.
  8. I suggest making figures with all the graphs shown, where they appear together, in order to be able to compare them in a better way.
  9. I am very sure that the article does not have the correct format from the journal, please make sure you have the format of the journal in each section, table, and every component.
  10. Review the entire article and avoid speaking in the first person. For example change "We found:" to "it was found". I couldn't specify to you exactly where the article was written in the first person because the article doesn't show the numbers on each line, please make sure you follow the proper format. 

Author Response

Dear Sir/Madam,

Thank you so much for the professional and comprehensive review of the manuscript. We are highly grateful for this in-depth and comprehensive review process which has made the manuscript better.

We sincerely hope that we have been able to answer your all the queries.

The manuscript entitled " Differential association of selected adipocytokines; adiponectin, leptin, resistin, visfatin, and chemerin, with the pathogenesis and progression of type 2 diabetes mellitus (T2DM) in Asir region of Saudi Arabia"  is focused on the measurement of 4 adipocytokines in patients diagnosticated with T2DM. Authors may find relevant suggested changes and recommendations:

  1. The authors must describe in more detail what is new in this manuscript. Please indicate how your work is different and important versus other studies published in the literature.

Response: Thank you very much for your comment. There are contradictory reports in literature about the association of various adipokines with diabetes and we haven’t come across any report in Asir region of Saudi Arabia. So, more and more original studies add to the available knowledge on this aspect and may offer more clarity in future analyses

  1. What are the advantages, novelty, or new contributions to the knowledge compared with other works reported previously in literature? Please list it in a table and explain.

Response: Thank you very much for your comment.    We found significantly decreased levels of serum adiponectin in T2DM patients as compared to control group, the decrease being more pronounced in obese and severely obese T2DM patients. We also found significantly higher serum leptin in females as compared to males in controls as well as all the four groups of T2DM patients. In male T2DM patients, a progressive increase was observed in leptin levels with increasing BMI, reaching to significantly altered levels, only in obese and severely obese patients. Serum leptin levels were also very significantly higher in severely obese female patients as compared to controls and patients with normal BMI and overweight patients.  No significant differences in serum resistin levels between males and females in controls as well as the T2DM groups irrespective of the BMI status in T2DM patients. We found no significant gender-based differences in visfatin levels, however significantly higher levels of visfatin were observed in T2DM patients irrespective of the level of obesity, although the higher values were observed in obese and highly obese patients.  We did not significant gender-based differences in serum chemerin levels in controls as well as in T2DM patients but in T2DM patients, chemerin levels showed a progressive increase with the increase in BMI reaching highly significant levels in obese and severely obese patients respectively. 

The subtle details are discussed pointwise in results and discussion sections.

  1. How do you select the N (number of participants) in each group? Describe the methodology in the proper section.  

Response: Thank you very much for your comment. Actually, our study was dependent upon the number of kits available to us. We had two kits of 96 test for each adipokine, one for patients and one for controls. After calibration and use of controls each ELISA kit generally is sufficient for 85-88 test samples. That is the reason for this limited number of samples. We have mentioned it as a limitation of the study in the conclusion.

  1. Table 1 is difficult to follow. I suggest splitting the table into 2 tables 1 about the biochemical parameters, and a second table about the demographic and anthropometric parameters.

Response: Thank you very much for your comment.     The needful has been done and is reflected in the manuscript as Tables 1a and 1b.

  1. Table 2 is difficult to follow too and is difficult to compare with Table 1. Please, be sure to rearrange the information so that it can be compared (men and women) and divide it into the characteristics that are related.

Response: Thank you very much for your comment.    The needful has been done and is reflected in the manuscript as Tables 2a and 2b.

  1. Table 3 should be a Figure in order to make it easier to follow and compare.

Response: Thank you very much for your comment.    The aim to this table is to offer the quantitative values for comparison. Figure is also present.  

  1. The colors are not suitable for a science project. Please use more formal but different colors that allow treatments to be compared.

Response: Thank you very much for your comment.    The needful has been done in news figures with “error bars” as suggested by another reviewer too. 

  1. I suggest making figures with all the graphs shown, where they appear together, in order to be able to compare them in a better way.

Response: Thank you very much for your comment.   The needful been considered. 

  1. I am very sure that the article does not have the correct format from the journal, please make sure you have the format of the journal in each section, table, and every component.

Response: Thank you very much for your comment.     The observation has been well considered and needful done.

  1. Review the entire article and avoid speaking in the first person. For example change "We found:" to "it was found". I couldn't specify to you exactly where the article was written in the first person because the article doesn't show the numbers on each line, please make sure you follow the proper format. 

Response: Thank you very much for your comment. The necessary modification s have been made as are reflected in the revised doc.

Reviewer 2 Report

In my opinion, the manuscript was prepared correctly. However, it requires minor changes

  • The Introduction Section explains the design of the study. The Authors well justify the research topic.
  • The Descriptions of the results were correct. However, it is necessary to organize the data in Tables 1 and 2. In addition, descriptions of individual data and the layout in particular fields require improvement. Nevertheless, the presented figures were prepared precisely and also legible.
  • In my opinion, the summary presented in the points in the Conclusions chapter is not necessary. Instead, this chapter should contain the most important conclusions of the study's authors.

Author Response

Dear Sir/Madam,

Thank you so much for the professional and comprehensive review of the manuscript. We are highly grateful for this in-depth and comprehensive review process which has made the manuscript better.

We sincerely hope that we have been able to answer your all the queries.

Thank you very much once again.

In my opinion, the manuscript was prepared correctly. However, it requires minor changes

Response: Thank you very much for your encouragement.   The needful has been done for each point.

  • The Introduction Section explains the design of the study. The Authors well justify the research topic.

Response: Thank you very much for your comment.    

  • The Descriptions of the results were correct. However, it is necessary to organize the data in Tables 1 and 2. In addition, descriptions of individual data and the layout in particular fields require improvement. Nevertheless, the presented figures were prepared precisely and also legible.
  • Response: Thank you very much for your comment. The necessary changes in the tables and figures have been made.
  • In my opinion, the summary presented in the points in the Conclusions chapter is not necessary. Instead, this chapter should contain the most important conclusions of the study's authors.
  • Response: Thank you very much for your comment.  Needful is done.

Reviewer 3 Report

This article is devoted to the study of association of different chosen adipocytokines (adiponectin, leptin, resistin, visfatin and chemerin ) with the pathogenesis and progression of type 2 diabetes mellitus (T2DM). The authors found a decrease in levels of adiponectin, an increase in levels of leptin, visfatin, chemerin, and, an absence of change in resistin levels  in T2DM patients (in case of chemerin - in obese and severely obese patients) in comparison with healthy control patients without T2DM.

Overall, Methods section is quite detailed, Results are described  and discussed properly.

Comments:

English language should be checked and corrected throughout the manuscript. For example, in the abstract section there is a sentence requiring a correction: “Resistin seems to unrelated to the insulin resistance in our set of patients.” 

“…we conclude that the altered secretions of four adipokines; adiponectin, leptin, visfatin and chemerin may contribute to defective production/action of insulin and concomitant insulin resistance.”

“Some reports have suggested that adiponectin can be considered as a biomarker for metabolic syndrome than for insulin resistance [64]. ”

Figure 3 legend should be checked as well.

How do the authors justify the choice of adipocytokines for their research among other adipocytokines? Why apelin, and ghrelin were omitted, for example?

In Methods section it should be described how T2DM was diagnosed in patients studied (what were the clinical criteria for the diagnosis, and which analyses were taken into consideration, for example, in order to distinguish cases from type 1 diabetes mellitus)?

It would be nice to have the data on medication taken by the patients studied since it could potentially interfere with the results interpretation.

Error bars in figure 1 would be helpful.

Also, the exact type of statistical analysis applied to results in each Table or Figure should be mentioned in a legend for that particular Table/Figure.

Results section.

“In females, Group B and D in T2DM patients showed significant differences in visfatin levels as compared to controls with p< 0.05 whereas T2DM patients in group C in T2DM showed very significant differences in visfatin… ” 

The statistical significance for the difference of Group B with the control is not shown in Table 4. Also, the symbol “#”, related to the comparison with groups A and B, and mentioned at the bottom of Table 4 is not shown in the table.

The paper can be accepted after minor revision related to answers to above mentioned questions.

Author Response

Dear Sir/Madam,

Thank you so much for the professional and comprehensive review of the manuscript. We are highly grateful for this in-depth and comprehensive review process which has made the manuscript better.

We sincerely hope that we have been able to answer your all the queries.

Thank you very much once again.

This article is devoted to the study of association of different chosen adipocytokines (adiponectin, leptin, resistin, visfatin and chemerin) with the pathogenesis and progression of type 2 diabetes mellitus (T2DM). The authors found a decrease in levels of adiponectin, an increase in levels of leptin, visfatin, chemerin, and, an absence of change in resistin levels  in T2DM patients (in case of chemerin - in obese and severely obese patients) in comparison with healthy control patients without T2DM.

Overall, Methods section is quite detailed, Results are described and discussed properly.

 Response: Thank you very much for your encouraging comments.

Comments:

English language should be checked and corrected throughout the manuscript. For example, in the abstract section there is a sentence requiring a correction: “Resistin seems to unrelated to the insulin resistance in our set of patients.” 

Response: Thank you very much for your critical observation comment. The needful has been done in the abstract and elsewhere too.

“…we conclude that the altered secretions of four adipokines; adiponectin, leptin, visfatin and chemerin may contribute to defective production/action of insulin and concomitant insulin resistance.”

Response: Thank you very much for your comment. The needful has been done.

“Some reports have suggested that adiponectin can be considered as a biomarker for metabolic syndrome than for insulin resistance [64]”.

Response: Thank you very much for your comment. The needful has been done in discussion.

Figure 3 legend should be checked as well.

Response: Thank you very much for your comment. The needful has been done as can be seen in the revised Ms.

How do the authors justify the choice of adipocytokines for their research among other adipocytokines? Why apelin, and ghrelin were omitted, for example?

Response: Thank you very much for your valid observation and comment.  There was no special reason. Our main limitation was the availability of resources. That why we included only small number of study subjects and only selected adipokines. We will keep this in mind for our future studies.

In Methods section it should be described how T2DM was diagnosed in patients studied (what were the clinical criteria for the diagnosis, and which analyses were taken into consideration, for example, in order to distinguish cases from type 1 diabetes mellitus)?

Response: Thank you very much for your comment. The needful has been done.

 It would be nice to have the data on medication taken by the patients studied since it could potentially interfere with the results interpretation.

Response: Thank you very much for your highly valid comment.  Most of our patients were on medication but we have missed that. As rightly pointed It could have made the data interpretation interesting and more meaningful. We will add this too as a limitation of our study and keep this important point for our future endeavors.

Error bars in figure 1 would be helpful.

Response: Thank you very much for your comment. The needful has been done.

Also, the exact type of statistical analysis applied to results in each Table or Figure should be mentioned in a legend for that particular Table/Figure.

 Response: Thank you very much for your comment. The needful has been done.

 Results section.

“In females, Group B and D in T2DM patients showed significant differences in visfatin levels as compared to controls with p< 0.05 whereas T2DM patients in group C in T2DM showed very significant differences in visfatin… ” 

The statistical significance for the difference of Group B with the control is not shown in Table 4. Also, the symbol “#”, related to the comparison with groups A and B, and mentioned at the bottom of Table 4 is not shown in the table.

Response: Thank you very much for your critical observation and comment. The needful has been done.

The paper can be accepted after minor revision related to answers to above mentioned questions.

Response: Thank you very much for your time and critical observation. Your valuable inputs have made this manuscript much better  

Reviewer 4 Report

dear authors, 

thank you for the paper. 

Indicate the study’s design in the title or the abstract, case-control study. 

The abstract is not informative and needs to focus on details of important results. 

The scientific background and rationale for the investigation are not clear. Some suggestions to read and consider adding: 

Díaz-Soto G, de Luis DA, Conde-Vicente R, Izaola-Jauregui O, Ramos C, Romero E. Beneficial effects of liraglutide on adipocytokines, insulin sensitivity parameters and cardiovascular risk biomarkers in patients with Type 2 diabetes: a prospective study. Diabetes Res Clin Pract. 2014 Apr;104(1):92-6. doi: 10.1016/j.diabres.2014.01.019. Epub 2014 Jan 25. PMID: 24530118.

Wang LK, Wang H, Wu XL, Shi L, Yang RM, Wang YC. Relationships among resistin, adiponectin, and leptin and microvascular complications in patients with type 2 diabetes mellitus. J Int Med Res. 2020 Apr;48(4):300060519870407. doi: 10.1177/0300060519870407. Epub 2019 Dec 31. PMID: 31891278; PMCID: PMC7607287.

González-López MA, Vilanova I, Ocejo-Viñals G, Arlegui R, Navarro I, Guiral S, Mata C, Pérez-Paredes MG, Portilla V, Corrales A, González-Vela MC, González-Gay MA, Blanco R, Hernández JL. Circulating levels of adiponectin, leptin, resistin and visfatin in non-diabetics patients with hidradenitis suppurativa. Arch Dermatol Res. 2020 Oct;312(8):595-600. doi: 10.1007/s00403-019-02018-4. Epub 2019 Nov 30. PMID: 31786710.

The statement "In the current study, we have studied the association of five
diverse adipokines; adiponectin, leptin, resistin, visfatin and chemerin with T2DM in a section of patients from Asir region of Saudi Arabia. " suggest an exploratory study. 

Describe the details of the setting, locations, and relevant dates, including periods of recruitment, exposure, follow-up, and data collection. 

Give the rationale for the choice of cases and controls. Why 1:1 ratio? Explain how the study size was arrived at and provide equations used for power calc. 

Describe all statistical methods, including those used to control for confounding  (was there gestational diabetes, Ramadan fasting (I think the period covers Ramadan), and any other confounding. Fasting could have affected results. 

Explain how the matching of cases and controls was addressed. 

results are too descriptive and offer little additional information - speak with statistician and results should be expressed as odds ratio and 95%CI. 

Figures need to include error bar. 

Report numbers of individuals at each stage of study—eg numbers potentially eligible, examined for eligibility, confirmed eligible, included in the study, completing follow-up, and analysed. Maybe draw flow chart. 

Results need to be adjusted and unadjusted. 

Summarise key results with reference to study objectives 

Discuss the generalizability - results cant be generalized to all pts with t2dm

Author Response

Dear Sir/Madam,

Thank you so much for the professional and comprehensive review of the manuscript. We are highly grateful for this in-depth and comprehensive review process which has made the manuscript better.

We sincerely hope that we have been able to answer your all the queries.

Thank you very much once again.

Indicate the study’s design in the title or the abstract, case-control study.

Response: Thank you very much for your comment. The needful has been done in the title

The abstract is not informative and needs to focus on details of important results.

Response: Thank you very much for your comment. Actually the details were avoided for the sake of brevity as our study includes five adipokines and word count in abstract is already more than 500.

The scientific background and rationale for the investigation are not clear. Some suggestions to read and consider adding: 

Díaz-Soto G, de Luis DA, Conde-Vicente R, Izaola-Jauregui O, Ramos C, Romero E. Beneficial effects of liraglutide on adipocytokines, insulin sensitivity parameters and cardiovascular risk biomarkers in patients with Type 2 diabetes: a prospective study. Diabetes Res Clin Pract. 2014 Apr;104(1):92-6. doi: 10.1016/j.diabres.2014.01.019. Epub 2014 Jan 25. PMID: 24530118.

Wang LK, Wang H, Wu XL, Shi L, Yang RM, Wang YC. Relationships among resistin, adiponectin, and leptin and microvascular complications in patients with type 2 diabetes mellitus. J Int Med Res. 2020 Apr;48(4):300060519870407. doi: 10.1177/0300060519870407. Epub 2019 Dec 31. PMID: 31891278; PMCID: PMC7607287.

González-López MA, Vilanova I, Ocejo-Viñals G, Arlegui R, Navarro I, Guiral S, Mata C, Pérez-Paredes MG, Portilla V, Corrales A, González-Vela MC, González-Gay MA, Blanco R, Hernández JL. Circulating levels of adiponectin, leptin, resistin and visfatin in non-diabetics patients with hidradenitis suppurativa. Arch Dermatol Res. 2020 Oct;312(8):595-600. doi: 10.1007/s00403-019-02018-4. Epub 2019 Nov 30. PMID: 31786710.

Response: Thank you very much for your comment. The needful has been done and one of the references has been included at S. No 22  

The statement "In the current study, we have studied the association of five
diverse adipokines; adiponectin, leptin, resistin, visfatin and chemerin with T2DM in a section of patients from Asir region of Saudi Arabia. " suggest an exploratory study.

Response: Thank you very much for your comment. The needful has been done at the end of introduction.

Describe the details of the setting, locations, and relevant dates, including periods of recruitment, exposure, follow-up, and data collection. 

Response: Thank you very much for your comment. The needful has been done in the “methodology section” and can be seen in the document with trach changes.

Give the rationale for the choice of cases and controls. Why 1:1 ratio? Explain how the study size was arrived at and provide equations used for power calc. 

Response: Thank you very much for your valid observation and comment.  There was no special reason. Our main limitation was the availability of resources. That why we included only small number of study subjects and only selected adipokines.  We had resources for only two ELISA kits (96 test each) for each adipokine and after adjusting for calibrators and controls, each of them was sufficient for 85-87 tests samples. That is why we took 87 T2DM patients and 85 controls. Nevertheless, we will keep this valid point in mind for our future studies.

Describe all statistical methods, including those used to control for confounding (was there gestational diabetes, Ramadan fasting (I think the period covers Ramadan), and any other confounding. Fasting could have affected results. 

Response: Thank you very much for your comment. The needful details have been added in the “methodology section”.  

Explain how the matching of cases and controls was addressed. 

Response: Thank you very much for your comment. The needful details have been added in the “methodology section”. We had initially recruited 100 controls (both males and females). After the initial screening, anthropometric details and biochemical parameters, only 85 healthy controls (44 males and 41 females) were included in the study. The age range for the controls was 28 to 62 years against that of 29 to 65 years for patient group.  The control group had waist to hip ratio of 0.86(0.84-0.95) and BMI of 21.78 ± 1.78 kg/m2.

 Figures need to include error bar. 

Response: Thank you very much for your valuable comment. The needful has been done for all figures.

Report numbers of individuals at each stage of study—eg numbers potentially eligible, examined for eligibility, confirmed eligible, included in the study, completing follow-up, and analysed. Maybe draw flow chart. 

Response: Thank you very much for your comment. The needful details have been added in the “methodology section”. We had initially recruited 100 T2DM (both males and females). After the initial screening, anthropometric details and biochemical parameters, only 87patients controls (46 males and 41 females) were included in the study. The rest of the patients who had either T1DM or has other associated chronic diseases were excluded.  For the controls we had initially recruited 100 subjects (both males and females). After the initial screening, anthropometric details and biochemical parameters, only 85 healthy controls (44 males and 41 females) were included in the study. The age range for the controls was 28 to 62 years against that of 29 to 65 years for patient group.  The control group had waist to hip ratio of 0.86(0.84-0.95) and BMI of 21.78 ± 1.78 kg/m2.

Results need to be adjusted and unadjusted. 

Response: Thank you very much for your valuable comment. The needful has been done in the results.

Summarize key results with reference to study objectives 

Thank you very much for your valuable comment. The needful has been done.

Discuss the generalizability - results can’t be generalized to all pts with t2dm

Thank you very much for your valuable comment. We agree 100% with your view as the number of study subjects is low, only a few selected adipokines were studied and many confounding factors like type and dosage of diabetic medications has not been taken into account. The information has been added in the conclusions.  

Round 2

Reviewer 1 Report

There is missing a Space in the abstract (44 males and 41 females)

Please review again your references.

About the question 

  1. What are the advantages, novelty, or new contributions to the knowledge compared with other works reported previously in literature? Please list it in a table and explain.

you Do NOT provide a table with the previous reports and your novelty.

Author Response

There is missing a Space in the abstract (44 males and 41 females)

Response:  Thank you very much for this critical observation.  Needful has been done.

Please review again your references.

Response:  Thank you very much for this critical observation and comment. Surely there were some issues which have been taken care of, as can be seen in the “track changes”.

About the question 

  1. What are the advantages, novelty, or new contributions to the knowledge compared with other works reported previously in literature? Please list it in a table and explain. You Do NOT provide a table with the previous reports and your novelty.

Response:  Thank you very much for the query. Kindly note that we had offered an explanation in our “previous response to comments” and the same points are discussed at respective places in different section of the manuscript; introduction, discussion and summary. We thought the explanation would be enough in this regard.

Nevertheless, if you need the same in tabular form, kindly suggest the contents and the format of the table. We will follow your instructions accordingly.

The extract from the previous response is reproduced as follows.

There are many contradictory reports in literature about the association of various adipokines with diabetes and we haven’t come across any report in Asir region of Saudi Arabia. As such this is the first report from this region of KSA on five adipokines. So, more and more original studies add to the available knowledge on this aspect and may offer more clarity in future analyses.

We found significantly decreased levels of serum adiponectin in T2DM patients as compared to control group, the decrease being more pronounced in obese and severely obese T2DM patients. We also found significantly higher serum leptin in females as compared to males in controls as well as all the four groups of T2DM patients. In male T2DM patients, a progressive increase was observed in leptin levels with increasing BMI, reaching to significantly altered levels, only in obese and severely obese patients. Serum leptin levels were also very significantly higher in severely obese female patients as compared to controls and patients with normal BMI and overweight patients.  No significant differences in serum resistin levels between males and females in controls as well as the T2DM groups irrespective of the BMI status in T2DM patients. We found no significant gender-based differences in visfatin levels, however significantly higher levels of visfatin were observed in T2DM patients irrespective of the level of obesity, although the higher values were observed in obese and highly obese patients.  We did not significant gender-based differences in serum chemerin levels in controls as well as in T2DM patients but in T2DM patients, chemerin levels showed a progressive increase with the increase in BMI reaching highly significant levels in obese and severely obese patients respectively. 

Reviewer 4 Report

thanks

Author Response

Moderate English changes required

Response:  Thank you very much for your comment. The linguistic check of the manuscript has been done by a faculty member of “English Department of University of Bisha”. The needful changes have been made and his contribution has been acknowledged at the end.